

# A major waterfall landscape maintained by fog drip water

**Lucheng Zhan[1,2], Jiansheng Chen [3,4], Chenming Zhang[5], Tao Wang[3], Ling Li[5] and Pei Xin[1,2]**

[1] State Key Laboratory of Hydrology-Water Resources and Hydraulic Engineering, Hohai University, Nanjing, 210098, China

[2] College of Water Conservancy and Hydropower Engineering, Hohai University, Nanjing, 210098, China

[3] Geotechnical Research Institute, College of Civil and Transportation Engineering, Hohai University, Nanjing, 210098, China

[4] College of Earth Sciences and Engineering, Hohai University, Nanjing, 210098, China

[5] School of Civil Engineering, the University of Queensland, St. Lucia, QLD 4072, Australia

*Correspondence to:* Jiansheng Chen (jschen@hhu.edu.cn) and Lucheng Zhan (luchengzhan@hotmail.com)





**Abstract**
The Chishui forest region in the southwest of China has a unique landscape with thousands of
waterfalls that produce a significant water yield even during and after a long dry period. However,
the sources of water for sustaining the waterfall landscape are poorly understood. We use stable
isotopes $^2$H and $^{18}$O to trace water in surface runoff and determine the runoff generation
mechanism in the catchments. Located on the pathway of water vapor from the neighboring
Sichuan Basin, the area is covered by a thick forest canopy above sandstone strata. The local
conditions combine to create a microclimate that favors formation of fogs at relatively high
frequencies. It was found that frequent fogs in this region act as a key water supplier for waterfalls
and play an important role in the regional hydrology. During the dry period starting from October,
waterfalls are mainly sustained by baseflow, 8-31% of which comes from frequent fog water
recharge. The waterfall landscape in the Chishui forest represents a unique characteristic of the
regional hydrological system in close connection with its geographical location, geology,
climatology and ecology.
**1 Introduction**
Many forest catchments experience a prolonged dry season with little or no rainfall, resulting in
significant surface water flow reduction and even dry riverbed in some cases (Jackson et al., 1995;
Liu et al., 2014; Querejeta et al., 2007). To adapt to the dry condition, local plants develop deep
roots, leaf withering and earlier flowering functions (Corbin et al., 2005; Goldstein et al., 2008).
In the southwest of China (Figure 1), there is a unique subtropical primeval forest called the
Chishui forest, where the vegetation coverage exceeds 90% with a wide range of species including
*Alsophila spinulosa* – a woody pteridophyte species surviving since the dinosaur age (Yang et al.,
2011). The forest is mainly underlain by Cretaceous or Jurassic quartz sandstone, which is different
from the surrounding areas (Qi et al., 2005). A large number of streams flow through the forest
region, resulting in the highest average drainage density (0.77 km/km$^2$) in China (Yang et al.,
2011). Located in the transition zone between the Yunnan-Guizhou Plateau and the Sichuan Basin,
this region is also characterized by a large number of faults and escarpments caused by past
geologic activities, which has led to the formation of thousands of waterfalls (Chen, 2003; Qi et
al., 2005). These waterfalls produce considerable water yield during the dry season when rainfall



is low (Chen, 2003). Another feature of the Chishui forest area is the frequent fog during all seasons and thus the lowest solar radiation level in China (Xu et al., 2002).

Fog water droplets in the air can be intercepted by plant leaves. These droplets then coalesce to form larger drops on the surface of the vegetation and fall to the ground. This process of water input is called fog interception or cloud water interception (Bruijnzeel et al., 2011; Prada et al., 2012). Previous studies showed that fog interception may affect significantly the hydrological cycle and ecology in tropical montane and coastal cloud forests around the world (Bruijnzeel, 2002; Bruijnzeel et al., 2011; Liu et al., 2014; Prada et al., 2016; Schmid et al., 2011). With more even distribution throughout the year than rainfall, fog precipitation has long been assumed as an essential additional water source in the relic laurel ecosystems of the Canary Islands (Aboal et al., 2000; García-Santos and Bruijnzeel, 2011; Ritter et al., 2008) and in the coastal forests of California (Dawson, 1998; Fischer et al., 2016; Ingraham and Matthews, 1990). The contribution of the fog water to local water budget and plant use cannot be overlooked in the ecosystems of tropical and temperate montane cloud forests as well as coastal ecosystems in major Mediterranean climate zones (Fischer et al., 2016). Once the fog water falls to the ground, it becomes an important water source for the forest ecosystem, contributing to soil water, aquifers and streams (Figueira et al., 2013; Hutley et al., 1997; Ingraham and Matthews, 1988; Prada et al., 2012, 2016). Since a conventional rain gauge is typically installed in the open field, it would not collect or record any amount of fog drip water that occurs under the forest canopy (Nagel, 1956; Vogelmann et al., 1968). The fog water's contribution is usually quantified indirectly, using methods such as artificial fog collection (Klemm et al., 2012; Ritter et al., 2008), throughfall measurement (Holder, 2004; Uehara and Kume, 2012) and modelling techniques (Imteaz et al., 2011; Ritter et al., 2008). The indirect methods rely on accurate rainfall and net precipitation measurements, which is difficult to achieve, making it a challenge to separate contributions by fog and rain (Schmid et al., 2011). Another problem is that the fog water contribution to the whole forest cannot be fully characterized by the water volume measurement at individual sites (Ritter et al., 2008). For example, Cavelier et al. (1996) measured fog water interception and rainfall at 14 stations in the montane forests across the Central Cordillera of western Panama, and found that fog drip water contributed between 2.4 and 60.6% of the total water input, subjected to a large degree of uncertainty due to changes of altitude and exposure to the prevailing winds. To better estimate the fog water contribution to local water budget, a better understanding of fog water-related





hydrological processes and a process-based quantification of fog water in groundwater and/or
surface water systems within the whole forest catchment is needed.

74        The fog water and rainwater exist in the hydrological system typically in a mixed form, which

makes it difficult to distinguish the two and trace them separately using traditional methods. As
the "fingerprint" of natural water, $^2$H and $^{18}$O stable isotopes have been widely applied to identify
different water sources in the hydrological cycle (Chen et al., 2014; Palacio et al., 2014; Zhan et
al., 2016). Due to different condensation conditions and processes, fog water and rainwater are
usually characterized by different isotopic compositions (Dawson, 1998; Gonfiantini and
Longinelli, 1962; Prada et al., 2016; Scholl et al., 2011). This offers an opportunity to distinguish
the fog water component in ecohydrological processes, and to quantify the ecological importance
of fog water to vegetation and its contribution to water budget in the cloud forests (Liu et al., 2014;
Schmid et al., 2011; Scholl et al., 2011; Zhan et al., 2017). Ingraham and Matthews (1988) used
isotopic methods to trace fog in northern Kenya and suggested that stable isotopes provided the
best tool available for tracing fog water movement into the groundwater system.  Comparisons
between the results of a mass balance model based on stable isotopes and direct fog deposition
measurements indicated that the former method provides a good estimation of fog interception
under a wide range of conditions (Schmid et al., 2011).

89        However, it remains a question whether fog can provide water for thousands of waterfalls in

a subtropical inland area such as the Chishui forest. The forest region is underlain by red sandstone,
which is different from the surrounding areas. As a subtropical cloud forest, the Chishui forest may
also be affected significantly by the fog water input just like many coastal cloud forests around the
world (Bruijnzeel et al., 2011). However, the link of fog water to the forest's unique waterfall
landscape and underlying hydrological processes are not well understood and require further
investigation. The annual rainfall in the area is not significantly different from the surrounding
areas (Figure S4 in supporting information). It is unclear why frequent fog appears in this region
and where the water through the large number of waterfalls in the dry period originates. To answer
these questions, several field investigations and water sampling campaigns over different temporal
and spatial scales were conducted in the study area.  Water samples for rainfall, fog interception,
springs and surface runoff within the study area were collected and analyzed for $^2$H and $^{18}$O isotope
compositions. Using the methods of isotope hydrology, the water input from fog interception in



the Chishui forest region was examined and the underlying hydrological processes were analyzed,
based on which an overall conservative estimation of fog water contribution was derived.

**2 Materials and Methods**

**2.1 Study site**

Located in the northeast of Guizhou Province, southwest China, the Chishui forest lies in the
transition zone between the Yunnan-Guizhou Plateau and the Sichuan Basin (Figure 1a). It is one
of the best preserved mid-subtropical plant communities of China and is home for China's biggest
communities of *Alsophila spinulosa* (Figure 2f), a relict woody pteridophyte species which has
survived over the time since the Mesozoic (Xu et al., 2002; Yang et al., 2011). Characterized by
the subtropical humid climate, this region has an average annual rainfall of 1047mm, 80% of which
falls between April and September. The average relative humidity is about 82% and the mean
annual temperature is 18.1 ℃, with 340-350 frost-free days per year. Because of frequent fog
events, the average sunshine time in the area is only 948.5-1292.5 h per year, making the local
solar radiation level the lowest in China (Xu et al., 2002).
Most areas of the Guizhou province belong to the karst landform dominated by limestone,
while the Chishui forest region developed on the larger and younger "Danxia landform" (red
sandstone uplands) in China (Qi et al., 2005). During the Jurassic period, the transition zone
between the Yunnan-Guizhou Plateau and the Sichuan Basin was a big inland lake surrounded by
limestone. Through tens of millions of years of sedimentation, red sandstone gradually covered
the bed of the lake. About 30 million years ago, following the Indian plate's subducting against
the Eurasian plate, the Yunnan-Guizhou region was gradually uplifted into a plateau, together with
the uplift of the sandstone formation in the lake area. Under the combined influence of hydraulic
erosion, gravitational collapses and weather denudation, high steep red cliffs gradually developed
along both sides of the stream valleys, forming a unique landform in the area (Qi et al., 2005; Li
et al., 2013a). Most of the exposed strata are sedimentary rocks, which are composed of Jurassic
and Cretaceous strata (99%), and quaternary strata (1%). The Jurassic and Cretaceous strata are
characterized mainly by lacustrine sediments such as red quartz sandstone or siltstone (Li et al.,
2013b) (Figure 2c and g). Geological activity in the junction of the Yunnan-Guizhou Plateau and
the Sichuan Basin results in a large number of faults and escarpments (Qi et al., 2005). Further



geological information about the study area can be obtained from the China Geological
Information Data Centre (http://geodata.ngac.cn/).
The study site within the Chishui forest region is a 2,390 km$^2$ catchment area (Figure 1c) near
the downstream reaches of the Chishui River, which is a tributary of the Yangtze River (Figure
1b). The elevation in the entire catchment decreases from about 1790 m on the southeast corner to
220 m on the northwest corner. This area has a highly developed surface runoff system, and covers
mainly the Datong and Fengxi subcatchments (Figure 1c), with the highest drainage density
reaching 1.37 km/km$^2$. Most waterfalls in the Chishui forest region concentrate in these two sub-
catchments. The Shizhangdong waterfall (Figure 2e), located in the upper reach of the Fengxi
stream, is the largest waterfall in the area and the whole Yangtze River basin. It is 80 m in height
and 76.2 m in width, and has a drainage area of only 170 km$^2$ but a discharge reaching 320 m$^3$/s
in the rainy season (Chen, 2003; Qi et al., 2005). The Datong catchment has the second largest
waterfall, the Sidonggou waterfall group, which has four levels with a maximum height of 50 m
and a maximum width of 40 m. The main waterfall area is densely forested, with vegetation
coverage greater than 95% (Figure 2d). Groundwater flows through clastic rock fissures in a
sandstone aquifer. Although the rainfall concentrates in the summer, the waterfalls still produce
considerable discharge during the dry period from October to March according to long-term field
observations by Chen (2003).

**2.2 Water sampling**

The first field investigation was carried out with surface water samples collected across the entire
Chishui River basin (Figure 1b) from June 5 to 12, 2011. Before this sampling work, the
southeastern region of China had experienced a serious drought that started in January 2011 and
lasted for nearly 5 months. Affected by this drought event, the rainfall amount from January to
May 2011 at a meteorological station near the Chishui forest was only 187 mm, which is the lowest
of the same period during 1981-2015 (Figure S1). Through the field investigation, we observed
obvious water shortage in stream flow within the upper reaches of the Chishui River because of
the drought (Figure 2a). There was no occurrence of fog in the upstream areas of high elevations.
In contrast, dense fog still occurred regularly in the Chishui forest region (downstream) in the early
morning, especially in the Datong and Fengxi catchments (Figure 2b). The ground was moist and
waterfalls flowed strongly, showing little impact by the drought (Figure 2b and c). Clean brown



5-ml vials with good airtightness were used for water sampling. Immediately after the water sample was taken, the vial was capped and tightly sealed with tape to minimize possible evaporation. In total, thirty-six water samples were collected during the first sampling campaign: ten from the mainstream of the Chishui River and twenty-six from the tributaries (Figure 1b). Twelve samples were taken in the forest area, including three waterfall samples. Geographical coordinates and elevations of all sample locations were recorded at the site using the global positioning system.

Another, more detailed field investigation with sampling focusing on the Chishui forest catchment (Figure 1c) was conducted from December 22 to 26, 2014 during the dry season. Sampling covered fog drips, springs, streams, waterfalls and rivers. Extensive sampling was conducted in the catchments of Datong and Fengxi streams, especially in three key waterfall landscape areas – Shizhangdong (Figure 1f), Sidonggou (Figure 1d) and Yanziyan (Figure 1e). Water samples of the mainstream and tributaries of the Chishui River in the forest catchment region were also taken. Every morning during the field trip, the forest was covered by fog (Figure 2d), which was gradually dispersed by sunshine later in the day. The sandstone wall under the canopy of the forest was found to be very moist (Figure 2g) and it could be easily seen that fog water kept dripping from the canopy to ground. Exposed rock surfaces were wetted everywhere by spring water flowing from sandstone fissures (Figure 2h). Waterfalls with different scales and shapes could be found on the cliffs of sandstone. Fog water samples were taken at different elevations by collecting water drops on the tips of plant leaves under the forest canopy using clean 5-ml vials (Figure 2i). There was no rain during the five sampling days (Figure S1) so that the water samples collected represented fog interception by the forest canopy. To prevent evaporation caused by increasing air temperature, fog water sampling was taken in early morning (before 9:00 am). Samples of spring water flowing from the sandstone fissures and surface water including streams, waterfalls and the Chishui River mainstream were also collected using 5-ml vials. In total, eighty water samples were collected during the second sampling campaign.

Sampling was also conducted from June to December 2015 to collect water samples twice a month from the Datong and Fengxi streams before they join the Chishui River (Figure 1c). The stream water samples were collected using 380-ml polyethylene bottles sealed and kept refrigerated at approximately 4 ℃. A rainwater collector was installed at an open site of the study area (Figure 1c, elevation: 265 m.a.s.l.) for collecting monthly rainfall water samples from January



2015 to December 2015. The collector was made of a 15-cm diameter polyethylene funnel draining
to a 10-L polyethylene bottle, containing a layer of oil film to minimize evaporation. At the end of
every month, the rainwater was sampled with a sealed 380-ml polyethylene bottle and then sent to
the laboratory for isotope analysis together with stream water samples. In total, 28 stream water
samples (14 for each stream) and 12 monthly rainwater samples were collected in 2015.
**2.3 Stable isotope analysis**
After each sampling campaign, water samples were immediately sent to Hohai University (Nanjing,
China) for the analysis of hydrogen and oxygen isotopes in the State Key Laboratory of
Hydrology-Water Resources and Hydraulic Engineering. The stable isotopic composition of
hydrogen was determined using an automated on-line elemental analyser (FlashEA HT) connected
to a Mat 253 mass spectrometer. This technique involved the reaction of sample water with carbon
at 1450 ℃ in a helium carrier gas. The product gases ($H_2$ and CO) were separated in a gas
chromatograph and analysed in the spectrometer for the hydrogen stable isotopic composition. For
the analysis of oxygen isotopic composition, water samples placed in vials were first flushed with
0.3% $CO_2$ for 10 minutes and then equilibrated with the 0.3% $CO_2$ headspace for 20 h at the
constant temperature of 25 ℃. Following the equilibration, vials were then inserted into a
GasBench Ⅱ system connected to the Mat 253 mass spectrometer. Hydrogen and oxygen isotopic
rates were reported in the standard δ-unit in parts per thousand with respect to the Vienna Standard
Mean Ocean Water. Analytical precisions were determined to be ± 2‰ and ± 0.1‰ for $\delta^2H$ and
$\delta^{18}O$, respectively.

212       The local meteoric water line (LMWL) in the study area was fitted using the monthly

precipitation isotope data. Data for daily rainfall in the sampling years, historical data (1981-2010)
for monthly average relative humidity, rainfall amount, daily temperature range, wind speed and
wind direction in the study area and surrounding regions were obtained from the China
Meteorological Database (http://data.cma.cn/). By comparing the isotopic compositions of
monthly precipitation from 2015 with the water samples collected during the three sampling
campaigns, key water sources and associated hydrological processes in the study area were
examined. Based on the analysis of rainfall-runoff process in the Datong and Fengxi catchments,
the two-compartment linear mixing model of Phillips and Gregg (2001) was used to estimate the
proportion (X) of fog water in the baseflow of the main waterfall catchments,



$$X = \frac{\delta_{baseflow} - \delta_{rain}}{\delta_{fog} - \delta_{rain}}$$

where $\delta_{baseflow}$, $\delta_{fog}$ and $\delta_{rain}$ are the isotopic values ($\delta^2H$ or $\delta^{18}O$) of baseflow, fog water and
rainwater, respectively.

**3 Results and discussion**


**3.1 Isotopic anomaly of the Chishui forest catchment discovered during the basin-scale**


**investigation**


Isotopic results (supporting information Database) of water samples collected across the entire
Chishui River basin in 2011 showed an enrichment of heavy isotopes in the mainstream of the
Chishui River from upstream to downstream (Figure 3b&c, section A-B-C-D as shown in Figure
3a). With decreasing elevation from 1455 m at No.1 to 226 m at No.12, the $\delta^2H$ and $\delta^{18}O$ values
of the mainstream increased from –55.6‰ and –8.63‰ to –41.1‰ and –6.32‰, with elevation
gradients of –1.18‰/100 m and –0.19‰/100 m, respectively. In the upstream catchment (A-B)
before the river flowed into the Chishui forest area, the isotopic values of the mainstream increased
slowly with the discharge of tributaries. Due to the altitude effect on isotopes in precipitation,
surface water from the tributaries became more enriched in heavy isotopes (Figure 3 c&d) with
the catchment elevation decreasing towards the downstream (Figure 3a). However, when the river
flowed through the forest region (B-D), its isotopic composition changed dramatically, showing a
slow depletion followed by a sudden enrichment. The isotopic depletion in section B-C might be
caused by input of local rainwater of depleted heavy isotopes due to the quickly increasing
elevation in the southeastern part of the forest catchment (Figure 1c). Further downstream, the
surface water in the Datong and Fengxi stream catchments (main waterfall region) became much
more enriched in heavy isotopes than that in the upstream areas (Figure 3c), leading to the
subsequent, rapid isotopic enrichment of mainstream from section C to D (No.11 to 12). This
phenomenon seems not consistent with the altitude effect of precipitation isotope because the
average elevation in the Datong and Fengxi catchments is higher than that of section B-C, with the
highest elevation reaching up to 1790 m, approximating that of the headstream of the Chishui
River. In this section (C-D), the elevation gradients for $\delta^2H$ and $\delta^{18}O$ in the mainstream reached –
29.0‰/100 m and –4.59‰/100 m, respectively, much greater than those of the entire mainstream





in the Chishui River basin (–1.18‰/100 m and –0.19‰/100 m). The large isotopic change after the river flowed through the main waterfall region indicated that surface water in the Datong and Fengxi catchments may have a water source of a different isotopic composition from that of local rainwater.

Based on the isotopic results of rainfall samples collected in the study area in 2015, the local meteoric water line (LMWL) in the Chishui forest was fitted as $\delta^2H = 8.65\delta^{18}O + 17.78$ (n=12, $r^2$=0.98) (supporting information Figure S2). As shown in Figure 3b, isotope values of surface water in the upstream Chishui River basin were scattered on or below the LMWL and formed SWL1 ($\delta^2H = 5.44\delta^{18}O - 7.29$, $r^2$=0.87) with a smaller slope than LMWL, indicating that it was sourced from rainfall and sometimes affected by evaporation. However, in the forest region, especially around the main waterfalls, isotope data were largely scattered above the LMWL along SWL2 ($\delta^2H = 4.58\delta^{18}O - 4.33$, $r^2$=0.80) significantly different from SWL1 (Figure 3b), indicating that the Datong and Fengxi stream catchments had a water source that is different from the water source in the upstream basin. Based on these basin scale isotope results and daily observations of fog formation and dripping during the field trip, we hypothesized that fog water might affect the isotopic composition of water in the waterfall areas. To verify this hypothesis, the second field investigation was carried out in December 2014 with a focus on the isotope characteristics in the Chishui forest catchment (Figure 1c).

## 3.2 Evidences for fog water recharge during the investigation focusing on the forest catchment

With $\delta^2H$ and $\delta^{18}O$ values ranging from 3.5‰ to 31.5‰ and –3.62‰ to 0.93‰ (supporting information Database), respectively, the fog water was much more enriched in heavy isotopes than the December rainwater (Figure 4), similar to the finding of previous studies on the isotopic composition of fog water (Corbin et al., 2005; Ingraham and Mark, 2000; Ingraham and Matthews, 1990; Scholl et al., 2011). However, the data points for all fog water samples appeared above the LMWL, which is different from the results in coastal areas where fog forms from water vapor coming directly from the ocean evaporation and hence has an isotope composition following the meteoric water line (Corbin et al., 2005; Ingraham and Matthews, 1990). Deuterium excess, defined as d-excess = $\delta^2H - 8\delta^{18}O$, can be used as an indicator of the origin of the water vapor (Liu et al., 2008; Prada et al., 2015). The fog water in the Chishui forest was characterized by an





average d-excess of 29.0‰, much higher than the global mean value (10‰) and that of the local rainwater in December (14.7‰). This indicates that the fog formed from condensed water vapor produced by different regional recycled meteoric water from evaporation (Froehlich et al., 2008; Liu et al., 2005, 2007). The fog water isotope line (FWL: $\delta^2H = 5.63\delta^{18}O + 26.24$, $r^2=0.78$; Figure 4) had a lower slope than that of LMWL, indicating that the fog water might have experienced evaporation. The fog droplets in the air have smaller sizes and higher surface/volume ratios than those of rain droplets, so they are more subjected to the evaporation effect despite relatively high local air humidity (Prada et al., 2015).

Consistent with the results from 2011, water samples collected from the mainstream of the Chishui River and the forest catchment in December 2014 showed significant differences in isotopic compositions (Figure 4), indicating again a different water input component in the forest area. The box plots showed similar variation ranges and average values of $\delta^{18}O$ and $\delta^2H$ among samples collected from waterfalls, streams and springs, indicating the linkage of the surface water (in waterfalls and streams) with spring water (groundwater). Water in the forest catchment showed little isotopic evidence for evaporation, which is consistent with the high humidity in the forest (supporting information Figure S3). Twelve of seventeen spring water samples, nineteen of twenty-three stream samples and twelve of fifteen waterfall samples showed isotopic characteristics above the LMWL. The d-excess values of waterfalls (15.0‰), streams (15.6‰) and springs (15.0‰) were higher than the volume-weighted mean value (11.8‰) of local rainfall and that of the Chishui river mainstream (11.9‰) (Figure 4d), indicating again an additional recycled water component contributing to water balance in the forest area. Rainwater in the rainy season (May to September) had d-excess values around 10‰ (supporting information Database), which could not explain the recycled water component in the surface water and groundwater. Rainwater in April was plotted above the LMWL and had a relatively high d-excess, but it was long before the actually sampling date and could not explain the similar findings in both June 2011 and December 2014. Although the data points of local rainfall isotopes in some winter months (November to January) also had higher d-excess values, the small rainfall amount (only 93 mm from November 2014 to January 2015) in these months would not be sufficient to generate significant runoff and affect the isotopic composition of surface water in streams and waterfalls. A more likely explanation for the higher isotopic values and d-excess of surface and groundwater in



the Chishui forest catchment, other than the rain-sourced water (Figure b, c&d), is the input to catchment from fog drip water in addition to local rainfall.

Through a review of 68 studies on a number of the world's mountain belts, Poage and Chamberlain (2001) concluded that there is a consistent linear relationship between isotopic values and corresponding elevations of water sourced from precipitation, with about 80% of the coefficients of determination ($r^2$ values) in these studies greater than 0.7. In the present study, fog water samples were collected at different elevations ranging from 250 m to 1140 m; but unlike precipitation, there was no correlation between fog water isotopic composition and elevation (Figure 5), indicating different isotope fractionation processes for rainwater and fog water condensation. Spring, stream and waterfall water samples were also collected at a wide range of elevations from 235 m to 1152 m (supporting information Database). However, the $r^2$ values for the linear regression of water isotopes with elevations for these samples were only 0.32 and 0.28 for $\delta^2$H and $\delta^{18}$O, respectively (Figure 5). Moreover, the fitted $\delta^{18}$O lapse rate in the Chishui forest region was only 0.1‰ /100 m, much lower than the rates for most regions of the world ~0.28‰ /100 m (Poage and Chamberlain, 2001). Since no elevation effect was found in the isotopic composition of fog water, the weak isotope-elevation correlation and small isotopic lapse rate in groundwater and surface water may be linked to the fog water input. The isotopic values the river mainstream increased quickly over the elevation range from 250 to 220 m where the mainstream received water from the Fengxi and Datong streams (Figure 5), indicating that fog water input happened mainly in these two subcatchments.

The two isotopic investigations in 2011 and 2014 revealed that surface water and groundwater (spring) in the forest region and upstream catchments were characterized by significantly different stable isotope compositions. The unusual isotopic characteristics in the forest, especially the waterfall-concentrated regions (Datong and Fengxi stream catchments), can be linked to the input of fog drip water, which also explained why the waterfall landscape remained wet even during drought conditions.

### 3.3 Rainfall-runoff process in the waterfall-concentrated catchments

The third sampling campaign was carried out to continuously monitor the isotopic compositions in the Datong and Fengxi streams (Figure 1c), which were fed by waterfalls in the corresponding catchments. According to the meteorological data (Figure 6), the annual precipitation at the study





site in 2015 was 1122 mm. The rain season in the region lasts from April to September, producing
nearly 80% of the total annual rainfall. The seasonal isotopic variation of the streams were much
smaller than that of the monthly rainfall (Figure 6). Although the Fengxi stream was more depleted
in heavy isotopes than the Datong stream in most months due to the higher altitudes in the Fengxi
catchment, both streams had a very similar temporal variation pattern in isotopic values. From
June to October, the isotopic compositions fluctuated significantly, but became relatively stable
from October to December. Streams usually consist of two hydrograph components: (1) surface
and near-surface quickflow in response to recent rainfall events, and (2) baseflow – water input
from persistent, slowly varying sources that maintains streamflow between rainfall events (Klaus
and McDonnell, 2013; Meyer, 2005; Muñoz-Villers and McDonnell, 2012). During the rainy
season, the rainfall rate was large enough to generate a considerable discharge of quickflow into
the Datong and Fengxi streams and cause obvious isotopic fluctuations of the stream water. When
the rainfall became small in the dry season, however, the streams in the catchment were maintained
by baseflow and displayed relatively stable isotopic compositions. The isotopic compositions of
the streams in December 2015 were similar to those in December 2014, indicating the stability of
isotopic compositions of baseflow. The isotopic compositions of the streams in June 2011 appeared
to be similar to those of base flow after a drought period in the area.

357        A phase lag of isotopic signal in the streamflow compared with the rainwater (input to the

catchment) can be found, indicating a transit time of rainwater in the catchment. From June to
early August, a depleting trend of isotopes $^2$H and $^{18}$O was observed in both the Datong stream and
Fengxi stream, corresponding to the overall isotopic depletion of rainfall from March to July with
a lag.  Similarly, stream isotopic values between early August and middle September showed a
delayed trend similar to that of rainwater isotopes from July to September. It seems that the lag
time between rainfall and streamflow had a trend of decreasing from the beginning of the rainy
season to the period from September to October, when isotopes of stream water and rainwater
varied almost simultaneously. From October to December, although the isotopes in rainwater
changed significantly, there was little variation in isotopes of stream water, indicating little direct
contribution of rain to the streamflow in the dry season. The rainy season in the Chishui forest
begins in April after a five-month long dry period. Because of the little rainfall in the dry season,
water stored in the unsaturated zone and aquifers greatly decrease over this period. It can be
assumed that at the beginning of the rainy season, the rainwater infiltrates to recover the soil





wetness conditions before generating runoff (Muñoz-Villers and McDonnell, 2012). As a
consequence, the observed delay time from rainfall to surface water was longer in the early stage
of the rainy season in the Chishui region. As the groundwater level, as well as the rainfall intensity
increased from June to October, rainwater could reach the streamflow more quickly. After October,
rainfall greatly decreased and became again a minor contributor to the streams.

The Chishui forest catchment is underlain by sandy soils and substrate of high permeability,
making it easier for rainwater to infiltrate and recharge the groundwater. Streamflow in the
montane catchments is usually primarily composed of near-surface runoff and baseflow of a longer
residence time. Typically the baseflow comprises a long-term mixture of rainfall, and its isotopic
composition can be estimated by the volume-weighted mean value of rainfall isotopes over a long
period (Goni, 2006). If rainfall is the only water source for the catchment, the isotopic composition
of baseflow should be similar to the volume weighted mean (VWM) isotopic value of annual
rainfall ($\delta^2H = -51.6‰$,$\delta^{18}O = -7.94‰$) or that of the wet season rainfall ($\delta^2H = -57.4‰$,$\delta^{18}O$
$= -8.55‰$) and the isotopic composition of stream water should fluctuate around this isotopic
value. However, as shown in Figure 6, stream water was more enriched in heavy isotopes than
VWM values in most months. This suggests that the baseflow in the Chishui forest catchments
was not just a mixture of rainwater from different rainfall events, but a mixture of both rainwater
and a considerable amount of fog drip water.

**3.4 Estimated contribution of fog water to the main waterfall catchments and comparison
with results from other areas around the world**

As discussed above, it is hypothesized that the baseflow in the Datong and Fengxi catchments is
likely a mixture of rainwater and fog drip water. Based on this hypothesis, the proportion (X) of
fog water in baseflow for each catchment can be estimated by the two-compartment linear mixing
model (Phillips and Gregg, 2001), with isotopic compositions of rainwater and fog water as two
end members. Isotopic values of Datong and Fengxi stream water samples collected in mid-
December 2015 were used to approximate the isotopic composition of baseflow in the
corresponding catchments. The isotopic composition of rainwater input was represented by the
volume weighted mean (VWM) isotopic values of monthly rainfall collected in 2015. Since fog
water samples were not collected over the whole year, the exact long-term isotopic input of fog
water could not be determined. In the mixing model, the $\delta^2H$ and $\delta^{18}O$ values of baseflow and



rainwater input were held constants while the values of each fog water sample were used in the calculation carried out for all fog water samples (10 in total for Datong and 8 for Fengxi). This resulted in a number of estimates of the fog water contribution (X) to baseflow according to the sample number for both catchments. The average, standard error and ranges of calculated fog water proportions are summarised in Table 1.

Fog water's proportions in baseflow calculated by $\delta^2H$ are similar to those by $\delta^{18}O$ as expected. Generally, the estimation shows that fog water accounts for a significant amount of the baseflow in the Datong and Fengxi catchments, with the proportion of 16-31% and 8-16%, respectively. Assuming that the proportion of fog water in the baseflow reflects the ratio of total annual fog water to total annual rainfall, we can estimate the annual fog water input to the area to be 98−504 mm in 2015 when the total annual rainfall was 1122 mm. The estimated fog water input in the Fengxi catchment appeared to be smaller than that in the Datong catchment. This difference may be caused by the simplification that two catchments have the same rainfall amount and rainfall isotopic input. In reality, the Fengxi catchment should have a higher rainfall rate with more depleted rainwater because of its higher altitude, and thus the proportion of fog water component in its baseflow should be higher.

The above analysis and estimates of the fog water contribution are subjected to uncertainties associated with a number of factors. First, the rainwater samples were collected in the study area with elevation of only 265 m.a.s.l., much lower than the Datong and Fengxi catchments. Because of the altitude effect on rainfall isotopes, the actual isotopic values of the rainfall end member (that should be used in the mixing model) are likely to be smaller than the values used for the analysis and estimation, which would lead to a higher fog water proportion. Secondly, the baseflow isotope values used in the analysis were based on the stream water samples collected in mid-December 2015 and the rainwater values based on the volume-weighted mean of rainfall data also from 2015; however the fog water data were collected from December 2014. Therefore, the real long-term isotopic input from fog water was not determined. The water vapor that was condensed to fog droplets mainly came from regional evapotranspiration, which originated from rainwater. The isotopic composition of fog water would vary seasonally like the rainwater (Scholl et al., 2011). In December 2014 when the fog water was collected, the rainwater was the most enriched in heavy isotopes in the year (Figure 6), which would have resulted in high isotope values for the collected fog water sample (higher than average). This would have led to underestimates of the overall fog



water contribution to the baseflow and the catchments. In short, while the estimates presented in Table 1 are subjected to uncertainties, they are likely to represent the lower bound of the fog water contribution to the Chishui forest region.

The estimated amount of fog water as part of the water balance in the study area is similar to those found in other fog-affected forests around the world. Table 2 lists the results of 13 studies (including the present one) quantifying fog water captured by forests using different approaches such as artificial fog collecting, throughfall measurement and modelling techniques. Located between 30° S - 41° N and 124° W - 152° E, most of these forests were found to have a general fog deposition rate of a few hundred millimetres per year. The amount of water produced by fog deposition depends partly on vegetation properties, climatic factors and terrain characteristics (Ritter et al., 2008). Although most of these forests are located in coastal areas, two of them (Hutley et al., 1997; Ritter et al., 2008) shown in Table 2 have the subtropical humid climate and are of the broad-leaved forest type, similar to the Chishui forest, and with annual fog deposition amount (450 mm and 251-281 mm) close to the estimation in this study.

## 3.5 Conceptual model of fog formation and recharge mechanism in the Chishui forest catchment

Composed of tiny condensed liquid water droplets suspended in the air, fog usually appears when water vapor becomes saturated as the temperature drops below the dew point. The formation of fog needs abundant water vapor and specific temperature conditions. The Sichuan Basin has a low altitude and high temperature, producing abundant water vapor through intense evaporation (Rong et al., 2012). With a relative flat topography, the southern edge of the basin provides a main pathway for water vapor transport out of the basin. The variation of seasonal prevailing wind direction (supporting information Figure S7) indicates that the study area receives water vapor from both Sichuan Basin and southeastern areas throughout the whole year. The river valley of the downstream Chishui River provides a natural channel for water vapor movement (Figure 7).

The outcropped strata are dominated by Jurassic mudstone, shale and limestone in the southern part of the Sichuan Basin, and Triassic limestone and dolomite in the northern part of the Yunnan-Guizhou Plateau. However, the study site is mainly underlain by Cretaceous or Jurassic quartz sandstone, different from surrounding areas (Figure 7). Compared with limestone and other rock types, sandstone has a greater water-holding capacity and weathers more quickly (Goudie et



al., 1970; Turkington and Paradise, 2005), which results in soils with better water and nutritional
conditions for local plant species to grow and survive (Jiang et al., 2012). As a result, this sandstone
area is mostly covered by forest, and has a different landform and microclimate from the
surrounding regions. When water vapor from surrounding areas moves towards the Chishui valley,
it is uplifted by the topography and cooled, resulting in fog. The forest's regulation on local
weather leads to stable air temperature and small wind speed (supporting information Figure S5 &
S6), both favoring the formation of forest fog. Usually bigger wind speed results in bigger fog
water interception (Prada et al., 2009, 2012), but the small wind speed is also benefit for the
occurrence and residence of frequent heavy fog. Moreover, compared with limestone, mudstone,
shale and dolomite, the quartz sandstone has a relatively high thermal conductivity and small
specific heat capacity (Clauser and Huenges, 1995; Kappelmeyer and Haenel, 1974). This allows
a relatively fast drop of near-surface temperature in the Chishui region after sunset, enhancing fog
formation.
Because of the special geographical location and geological conditions, the Chishui forest
region is covered by heavy fog in most days of the year. In arid regions such as northern Kenya,
the residents collect fog water for their water supply by installing cooling screens windward
(Ingraham and Matthews, 1988). The dense canopy of the Chishui forest, especially in the Datong
and Fengxi catchments, functions like a natural screen to capture fog water droplets from the air
and produce larger water drops that subsequently fall to the forest ground. Some of the fog water
is lost via evapotranspiration while the rest infiltrates into soils together with rainwater and
recharges local groundwater. Rainwater can saturate the soils and form near-surface flow and
quickly enter surface water in the rainy season with abundant and intense rainfall, but mainly
infiltrates and recharges groundwater that leads to baseflow in the dry period. Unlike rainwater,
fog water recharge is persistent and despite its relatively low intensity, leads to considerable water
input throughout the year.
Sedimentary sandstone in the Chishui forest region has obvious layering and is made up of
sand particles with different sizes (Figure 2c), causing certain differences in compactness and
permeability (Peng, 2001). When the groundwater stored in the sandstone (suspended aquifers)
comes across the argillaceous sandstone layers that have relatively smaller permeability, it will
move along these structures and finally flow out as spring water (Figure 2g). Such springs exist
throughout the valleys, forming large numbers of streams which become waterfalls at geological





escarpments. Although the rainfall amount in the study area is similar to the surrounding areas
(supporting information Figure S4), fog water provides considerable recharge to groundwater. As
a consequence, groundwater can maintain the large discharge of waterfalls during a long dry period.
The special geographical location, geological and lithologic characteristics, as well as the
vegetation conditions of the Chishui forest region together create this unique waterfall landscape.
How the area developed such a landscape under combined influences of various interacting factors
over time would be an interesting question for future research to better understand the interactions
among hydrological, geological and ecological processes in the evolution of an earth system.
**4 Concluding remarks**
Isotopic signatures of stream water, fog water and rainfall in the Chishui forest indicate that
frequent fog drip is a key water source for sustaining streamflow and plays an important role in
the regional ecohydrology. Since the forest is located in the vapor passage in the southern part of
the Sichuan Basin, abundant water vapor from the basin and southeastern regions provides an
essential condition for the fog formation. The special properties of sandstone strata create a thick
forest landscape and a microclimate that also favor the formation of frequent fogs in the study area.
Seven-month monitoring of the isotopes in streams fed by water from waterfalls in the catchments
indicates that surface water flow is formed by subsurface runoff with different proportions of
groundwater and near-surface flow in different time periods. During the dry period starting around
October, surface water that forms thousands of waterfalls is mainly sustained by baseflow, 8-31%
of which comes from the frequent fog water recharge. Fog is intercepted by the leaves of forest
plants and forms large water drops that fall to the ground, infiltrate the soil and recharge
groundwater together with rainwater. Groundwater flows out of sandstone layers and forms springs,
which then converge into streams and waterfalls. Overall, fog water provides 98-504 mm/a
recharge to the hydrologic system, in addition to the rainfall input (1122 mm in 2015).
The present study, based on the isotope method, has demonstrated that fog water contributes
significantly to the water balance and contributes to baseflow in the waterfall landscape in the
Chishui forest region. This unique waterfall landscape is inseparable from the geographical
location, geological activities, lithologic characters as well as the vegetation conditions. We
suggest that future investigations should be carried out with long-term measurements of isotopes
and hydrological parameters, including river flow, local rainfall and fog water dripping rate, to





further explore the processes underlying the behavior and evolution of this complex
ecohydrological system.

## Acknowledgments

Data used in this study are included in the supporting information. This research was funded by
the National Natural Science Foundation of China (51578212) and the National Basic Research
Program of China (2012CB417005). We gratefully acknowledge the funding from the China
Scholarship Council. We thank laboratory technician Zhiguo Su for the isotopic analysis of
samples. We also thank Shiyin Zhang, Jin Geng, Wencheng Tao, Hongbo Zhao for their assistance
in collecting field samples.

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




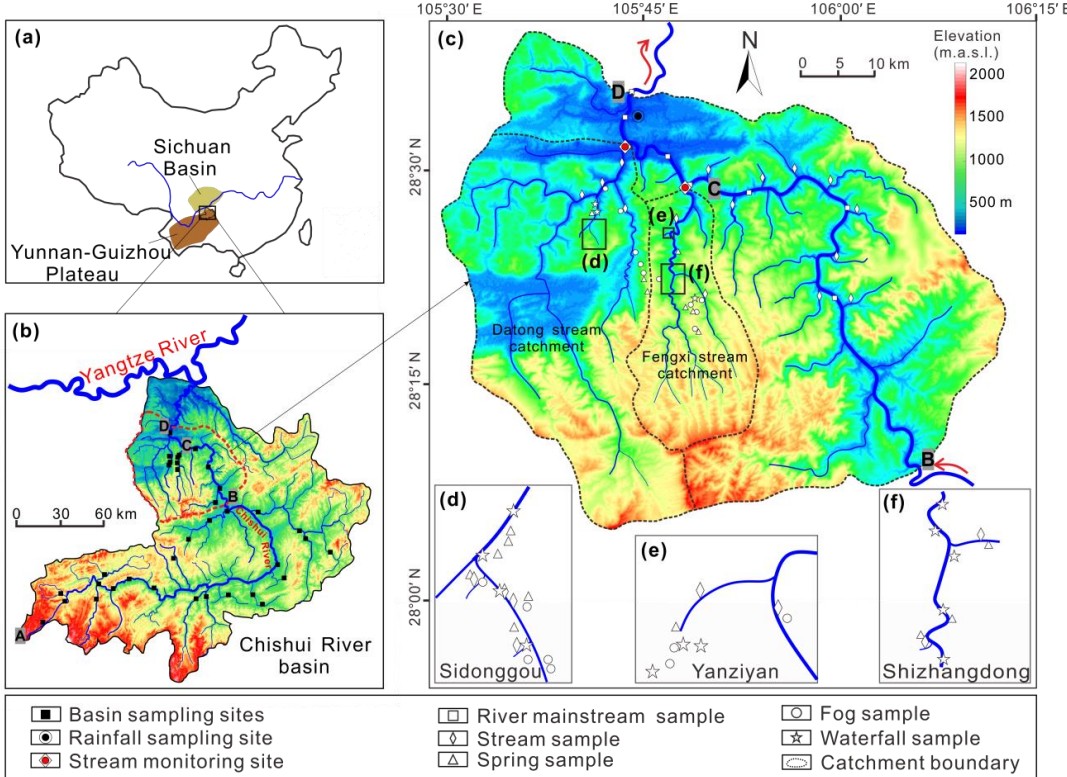


**Figure 1.** Location of the study area and sites for water sampling in the Chishui River basin and Chishui forest catchment. (**a**) Study area located in the transition zone between the Yunnan-Guizhou Plateau and the Sichuan Basin in the southwest of China. (**b**) Sampling sites (June 2011) in the Chishui River basin and the location of the Chishui forest catchment. (**c**) Sampling sites (December 2014) in the Chishui forest catchment, including the Datong and Fengxi stream catchments (main waterfall area). (**d**) Sampling sites in Sidonggou waterfall landscape area. (**e**) Sampling sites in Yanziyan waterfall landscape area. (**f**) Sampling sites in Shizhangdong waterfall landscape area. The digital elevation data are sourced from China Geospatial Data Cloud (http://www.gscloud.cn/).





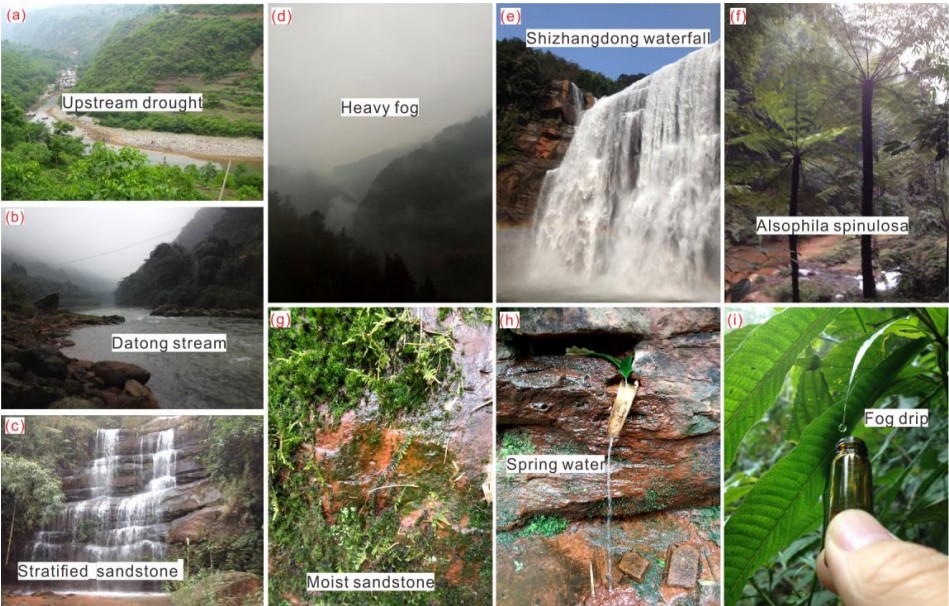

**Figure 2.** Pictures taken during the field investigation and sampling. (**a**) Small stream discharge in the upstream catchments of the Chishui River due to the drought in the spring of 2011. (**b**) A photo taken in June 2011, showing the fog occurrence and the large discharge in the Datong stream in contrast with the upstream situation shown in (a). (**c**) Photo of a small waterfall on the stratified sandstone taken during the first field investigation. (**d**) Heavy fog during the second field investigation in December 2014. (**e**) The biggest waterfall (105 °44'27.52" E, 28 °21'35.14" N) in the study area photographed in December 2014. (**f**) *Alsophila spinulosa*, an ancient species surviving since the dinosaur age, grows in the study area. (**g**) Red sandstone under the forest canopy wetted by fog water drops. (**h**) Spring water flowing out from the fissures of sandstone. (**i**) Sampling fog water by collecting water drops on the tips of plant leaves under the forest canopy.



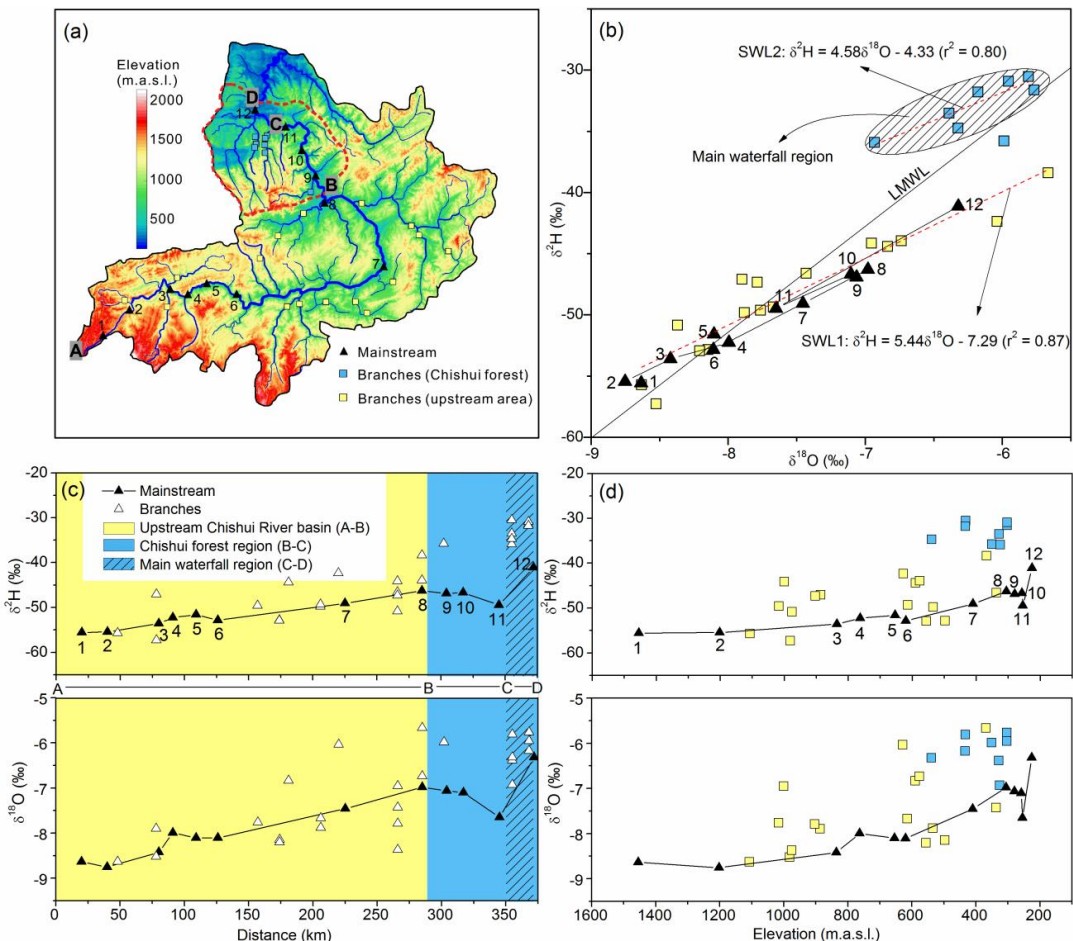

**Figure 3.** Isotopic characteristics of surface water in the Chishui River basin based on water samples collected in June 2011. (**a**) Sampling sites of the Chishui River mainstream (No.1 to 12) and its tributaries/branches. (**b**) The relationship between $\delta^{18}O$ and $\delta^{2}H$ values. (**c**) Isotopic variations of surface water along the Chishui River basin. (**d**) The relationship between isotopic values and sampling elevations. The local meteoric water line (LMWL) was fitted for monthly rainfall samples collected in 2015 (supporting information Figure S2). The x axis of (c) stands for the distance from the river headstream (A) to each sampling location along the mainstream or each junction of the mainstream and branch. SWL stands for surface water line.



**Figure 4.** Relationship between $\delta^{18}O$ and $\delta^2H$ (**a**) of all water samples collected in the Chishui forest catchment during the sampling campaign conducted in December 2014, and box plots of $\delta^2H$ (**b**), $\delta^{18}O$ (**c**) and d-excess (**d**) values for different water types. The sizes of dark grey circles in (**a**) represent the relative amount of rainfall in different months. FWL stands for the fog water line and VWM for the volume weighted mean.





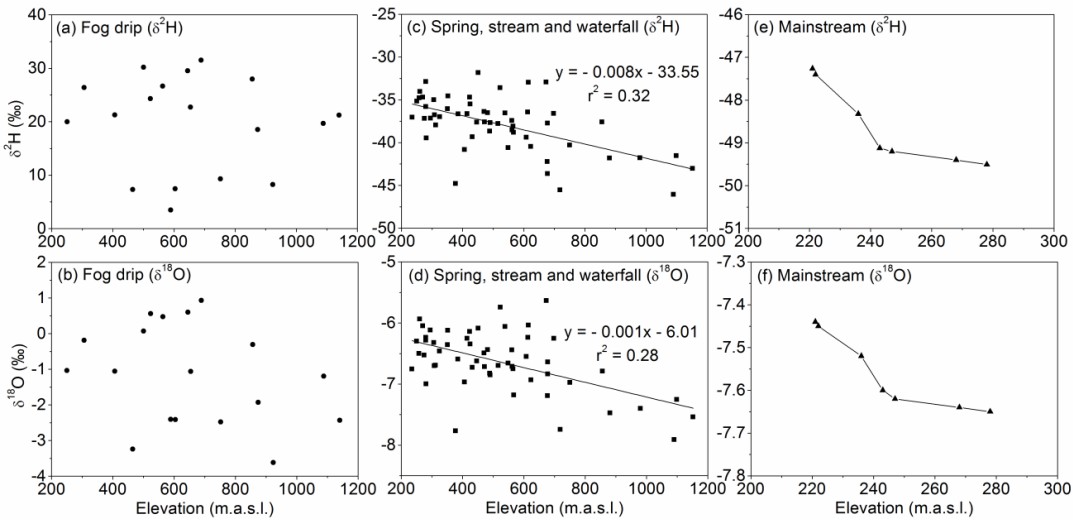


**Figure 5**. Relationships between isotopic values and corresponding elevations for fog drip, spring,
stream, waterfall, and the Chishui River mainstream water samples collected in December 2014.






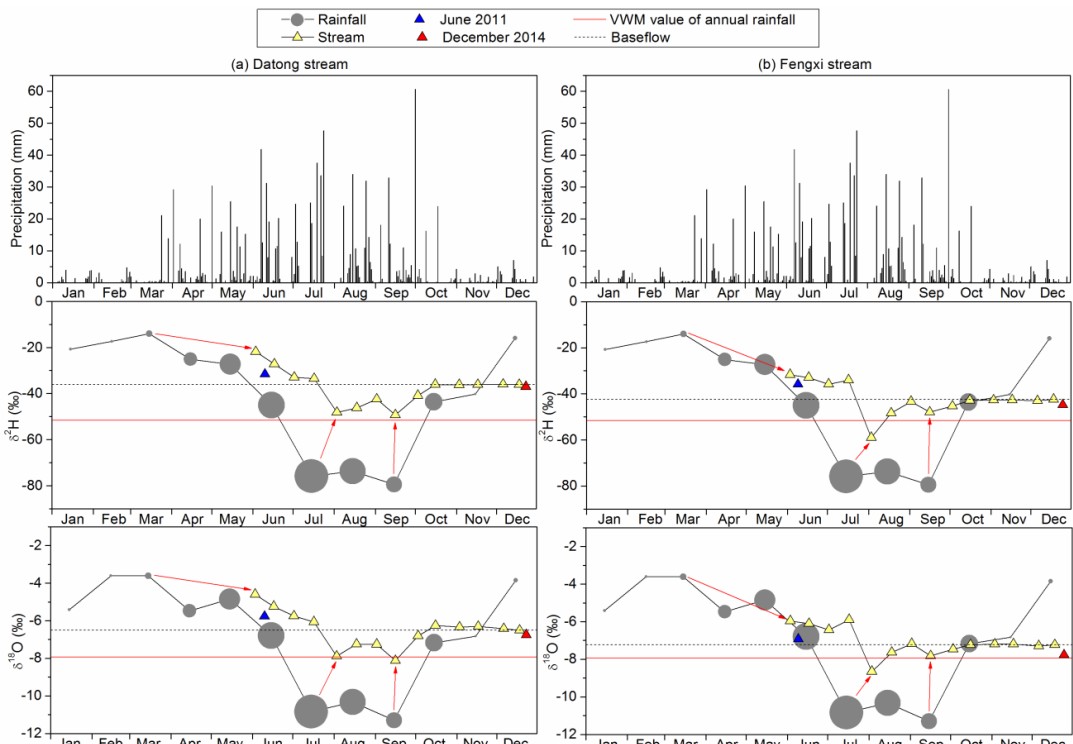

**Figure 6.** Comparison of monthly variations of isotopic compositions between precipitation and
water from Datong and Fengxi streams in 2015. Daily rainfall data at a nearby meteorological
station in the study area (Figure S3, about 35 km away from the Fengxi catchment) were obtained
from China Meteorological Database (http://data.cma.cn/). The sizes of dark grey circles represent
the relative amount of rainfall. The volume-weighted mean (VWM) isotopic values of monthly
rainfall were calculated using monthly rainfall isotopes and monthly rainfall amount in 2015. The
isotopic compositions of baseflow in the two catchments were correspondingly represented by the
isotopic values of stream samples collected in mid-December 2015.

754



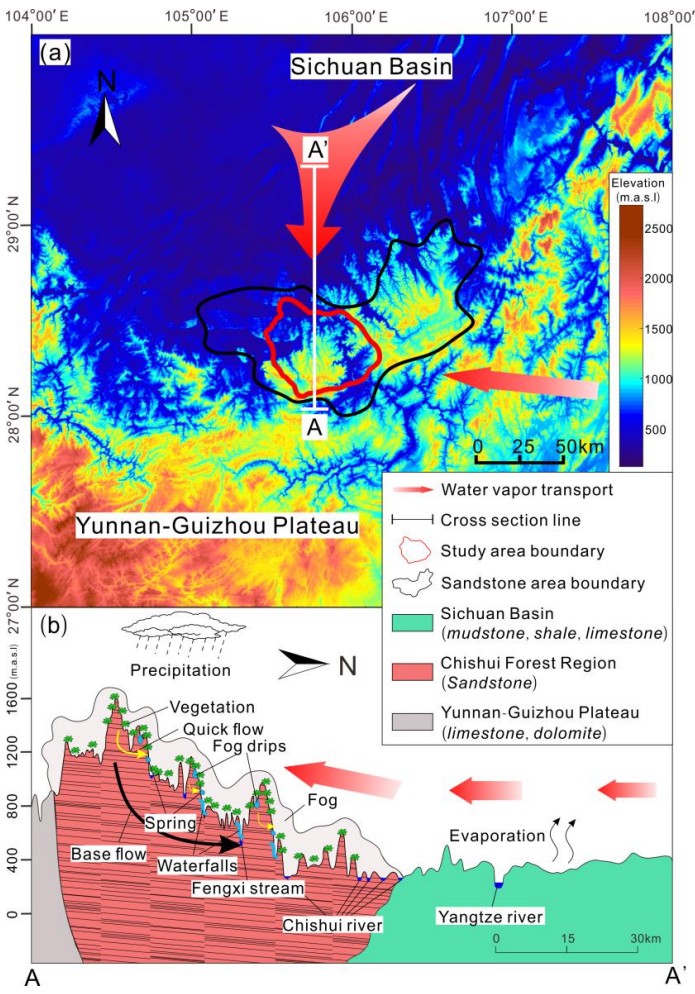

**Figure 7**. Map of the site and surrounding areas, and diagram of fog formation and hydrologic

process in the study site. (**a**) Digital elevation map of the transition zone between the Yunnan-

Guizhou Plateau and the Sichuan Basin, as well as the boundaries of Danxia (red sandstone)

landscape and study area. (**b**) A simplified geological cross-section (A-A') shown in (**a**) describing

the water vapor source for fogs and the concept model of fog recharge process in the study

catchment. The geological information is obtained from China Geological Information Data

Centre (http://geodata.ngac.cn/). The digital elevation data is from China Geospatial Data Cloud

(http://www.gscloud.cn/).





**Table 1 Isotopic compositions of the end members for the mixing model of baseflow and calculated**
**proportions of fog water input in Datong and Fengxi catchments.**

| Catchment | $\delta_{baseflow}$ (‰) | | $\delta_{rain\_VWM}$ (‰) | | $\delta_{fog}$ (‰) | | X (%) | |
|---|---|---|---|---|---|---|---|---|
| | $\delta^2H$ | $\delta^{18}O$ | $\delta^2H$ | $\delta^{18}O$ | $\delta^2H$ | $\delta^{18}O$ | Calculated by $\delta^2H$ | Calculated by $\delta^{18}O$ |
| Datong | −36.2 | −6.50 | −51.6 | −7.94 | 22.3±3.0 (3.5-31.5, N=10) | −0.61±0.43 (−3.24-0.93, N=10) | 21±1.0 (19-28) | 20±1.5 (16-31) |
| Fengxi | −42.4 | −7.23 | −51.6 | −7.94 | 16.6±2.5 (7.5-26.6, N=8) | −1.83±0.44 (−3.62-0.47, N=8) | 14±0.5 (12-16) | 12±0.8 (8-16) |

Note: The rainwater input was represented by the volume weighted mean (VWM) isotopic values of 12 monthly
rainfall collected in 2015. Mean values with standard errors for fog water isotopic composition and its proportions (X)
in the baseflow were shown, as well as the minimum and maximum (in parentheses). Detailed results for every fog
water sample can be found in the supporting information Database.





770          **Table 2 Estimation of fog deposition rate in foggy forests around the world.**

| Study | Location | Latitude | Longitude | Forest type | Annual rainfall (mm) | Annual fog water input (mm) | Fog proportion, X (%) |
|---|---|---|---|---|---|---|---|
| Cavelier et al., 1996 | Central Cordillera of Panama | 8°41'23"N | 82°11'28"W | Tropical montane forest | 3355-5759 | 142−2295 | 2.4-60.6 |
| Hutley et al., 1997 | Queensland, Australia | 28°13'55"S | 152°25'23"E | Subtropical rainforest | 1125 | 450 | 40 |
| Dawson, 1998 | Northern California | 41°33'N | 124°4'W | Costal redwood forest | 1315 | 224−447 | 17−34 |
| Holder, 2004 | Guatemala | 15°5'57"N | 90°3'59"W | Tropical montane cloud forest | — | 270 | — |
| Liu et al., 2004 | Xishuangbanna, China | 21°55'39"N | 101°15'55"E | Tropical rainforest | 1718 | 89.4 | 5 |
| Chang et al., 2006 | Taiwan | 24°35'N | 121°25'E | Mountainous coniferous forest | 2940 | 328 | 10 |
| Eugster et al., 2006* | Puerto Rico | 18°16'17"N | 65°45'39"E | Tropical montane cloud forest | — | 1591 | 13 |
| Del-Val et al., 2006 | Chile semiarid region | 30°40'S | 71°30'W | Costal rainforest | 147 | 200 | 58 |
| Ritter et al., 2008 | Canary Islands | 28°8'20"N | 17°15'25"W | Subtropical elfin laurel forest | 635−1088 | 251−281 | 21-28 |
| Prada et al., 2009 | Madeira Island | 32°45'37"N | 17°2'50"W | Costal forest | 1660 | 153.4 | 13 |
| Schmid et al., 2011* | Costa Rican | 10°21'33"N | 84°48'5"W | Montane cloud forest | — | 438 | 5−9 |
| Uehara and Kume, 2012* | Northern Japan | 36°33'58"N | 137°36'22"E | Alpine forest | — | 1226 | 35 |
| This study | Southeastern China | 28°21'35" N | 105°44'28" E | Montane cloud forest | 1122 | 98−504 | 8−31 |

— no result shown in the corresponding paper. Results of rainfall and fog water input in the studies are all converted to mm/year
for comparison, although the results of some studies (*) were only given for several months.