# Peer review of "Discussion started: 9 February 2018 © Author(s) 2018. CC BY 4.0 License."

_Hydrology and Earth System Sciences, 2018_

## Referee Comment (RC1) · Anonymous Referee #1 · 20 Mar 2018

Review for "A major waterfall landscape maintained by fog drip water" by Lucheng Zhan et al. Submitted to HESS

The manuscript by Zhan et al. investigates the influence of fog water on the streamflow of streams and water falls in the Chishui River Basin, in particular during the dry season. The investigation relies on the analysis of the isotopic composition of fog, stream, and rain water to calculate the proportional contribution of fog water input to the streamflow.

While the isotope analysis is very thorough and that I think the investigation might be an interesting addition to the literature showing the importance of fog water contribution to the water cycle, I think that a few key aspects of the analysis are presented without any data backing them up, making some of the arguments purely qualitative. As is, I

would not recommend this manuscript for publication in HESS.

Here, I list the main issues that I have found with the study:

1) Data on the amount and the timing of the fog events is clearly lacking in this study. Currently, fog events are described as happening "often" or "most days". However, in a place with such a strong seasonality in rainfall, one would also expect a seasonality in the fog formation. In particular, data on which days and for how long fog forms can be easily recorded using a simple camera and basic image analysis. I think this information is vital to get a full picture of the hydrology of the site, and this new information could potentially completely change the conclusion.

2) The geology of the area is invoked throughout the manuscript to explain the type of vegetation present, the occurrence of fog, and the large number of springs and waterfalls. However, no data is provided about the permeability of the different rock formations and details about the underlying soil and rock layers come from general geological information, as opposed to being new information provided by this study. As such, I think the geological argument should be shortened and condensed within the discussion section.

3) Organizational issues: I would suggest separating the results and discussion sections. As it is, it is currently very hard to understand how the different arguments work together towards the conclusion. I also think that the subsections in Section 3 could benefit from being further subdivided into subsubsections, making each piece of the argumentation clear. Finally, I am really confused as to why the figures are presented in an order that does not match their order of appearance in the text.

4) The English language needs to be improved. In the following, I do my best to correct small mistakes and point out sentences that need to be reworked.

Technical issues:

Abstract:

- L19 and L20: "fog", not "fogs"

Introduction:

- L29: dry conditions

- L42: "seasons and thus the. . ." change to "seasons, resulting in the. . ."

- L46: may significantly affect

- L48-49: Rephrase the sentence starting with "With more even. . ."

- L49: has long been assumed to be

- L52 to 55: rephrase the sentence starting with "The contribution of the. . .."

- L63: measurements, which are

- L64 contributions from fog and rain

- L69: total water input, and was subject

- L89: Maybe replace the beginning of the sentence by: "A question remains as to whether or not fog can. . ."

- L91: the surrounding area

- L92: also be significantly affected

- L93: the link between fog water and the forest's unique

- L94: the link [. . .] underlying hydrological processes is not well understood and requires further

- L96-97: rephrase the sentence starting with "It is unclear. . ."

- L101: remove "Using the methods of isotope hydrology"

Section 2: Materials and Methods

- L116-132: Remove. See my comment above: the argument based on the geology of the area should be condensed and moved to the discussion.

- L140: It is 80m high

- L141: 76.2m wide

- L142: during the rainy season

- L153: remove "affected by this drought event

- L153: the total rainfall from January

- L154: Name and exact geolocation of the met station??

- L154-155: which was the lowest record for that period between 1981 and 2015.

- L155: During fieldwork, we observed

- L156: quantify "obvious"

- L156: replace "water shortage in stream flow" by "a decrease in stream flow"

- L160: showing little impact of the drought

- L161: add details on the brand and the cap/septa type

- L168: A second, more detailed

- L170: streams, waterfalls, and rivers

- L174: during fieldwork

- L176: rephrase or remove "very moist" and "easily seen"

- L178: what do you mean by the "scale" of a waterfall? Do you mean its size?

- L179: sandstone cliffs

- L190: what's an "open site"? Do you mean with full view of the sky (and not under the

canopy?)

- L198-211: You need to give more details about the actual analysis: number of standard used and in which position, post-processing, etc. . . Also, please explicitly describe the brand, name, and isotope value of the standards used.

- L201: in-line

- L202: this technique involves

- L203: the product gases are separated

- L210: 2permil for 2H on a mass spec is really high. Can you comment on why this value is so high?

- L212: Is the LMWL built from the rainfall data you collected and that is cited later on? Or is it from previous studies? Either way, please explain how your LMWL was built.

- L213-216: Please rephrase the sentence starting with: "Data for daily rainfall. . ."

- L219: What analysis of rainfall-runoff are you referring to?

Section 3: Results and discussion

Estimates of the error in your measurements is missing throughout this entire section.

- L228: isotopic composition

- Section 3.1: this entire subsection is based on the qualitative analysis of Figure 3. A lot of the conclusions derived are based on two or three data points. Overall, I think that figure 3 and Section 3.1 end up undermining the analysis. Either quantify the "dramatic changes" (L238, L244) or simply remove the figure and section, since I think the following sections/figures are a lot more convincing.

- L254: isotopic composition

- L273: Maybe Kaseke et al (2017) could be a good, more recent reference to

add to this list Kaseke, K. F., Wang, L., & Seely, M. K. (2017). Nonrainfall water origins and formation mechanisms. Science Advances, 3(3), e1603131–. http://doi.org/10.1126/sciadv.1603131

- L286: How about evaporation effects once the water deposits on the leaf?

- L312: Here would be a good place to have a subsubsection

- L314: "water sourced from precipitation": what water? Soil water, stream water? Please rephrase.

- L323: average rate?

- L326-329: Please rephrase the sentence starting with "the isotopic values of the river..."

- L330-335: Remove, not useful

Section 3.3: here again, there is a real need to provide a more rigorous quantification of the various arguments presented.

- L352: during the dry season

- L353: quantify the variation in stable isotope composition during the dry season

- L353-354: the isotopic composition of the streams in December was similar to that of ... the isotopic composition of the baseflow.

- L357: Here, I am not convinced that what is seen in Figure 6 is really a lag in the isotopic composition of stream compared to rain water. In particular, the lack of stream data until May makes it difficult to properly compared both time series. In addition, the difference in sampling time between the rain and the stream water analysis means that you might be missing details in the variation of rain water composition, especially during the rainy summer months.

- L363: decreasing trend
- L370: infiltrates and recovers soil saturation

- L373: what do you mean by "as the groundwater level"?

- L376: quantify "high permeability

- L377: easy, not easier

- L377: in montane catchments

- L385: How about the effect of soil water evaporation, which would have a similar effect of enriching rainwater before it enters streams?

- L406-445: Here is where additional information on fog events would be really helpful to solidify the argument.

- L417: Potential break for a subsubsection

- L428: vary seasonally: data?

- L431: an underestimation of the overall fog

- L443: a subtropical humid climate

Section 3.5: here is where you could condense all the geological information + include more specific information regarding the different rock formations, in particular permeability.

- L476: "most days of the year": please show data on fog formation timing!

- L476-478: remove

- L481: into soil together

- L483: intense rainfall. However, most rainwater infiltrates

- L485: fog water is persistent: is it? Again, having fog formation data is crucial to this argument.

Figure 3: See my comment above, I do not think that this figure helps the analysis and it should be removed or moved to the supplemental information

Figure 4: this is a great summary figure of the isotope data

Figure 6: Describe the figure from top to bottom, starting with the rainfall data. Explicitly mention "top", "center", and "bottom". Why is the stream data before May not available?

Table 1: it would probably be enough to simply give the average and the STE, and not the range. Add one extra column for the number of samples.

Figure S1: sampling campaign was conducted from. . .. Rainfall data from the China Meteorological. . .

Figure S3: the digital elevation data for Figures S3 to S7, not S2 to S6.

Figure S7: Are the differences in wind direction between the 4 maps actually significant?

---

## Referee Comment (RC2) · Anonymous Referee #2 · 21 May 2018

This paper analyses isotopic data collected in the Chishui forest region in the southwest of China to characterise the underlying hydrological processes and quantify the contribution of fog water to a waterfall system. By using data collected in two campaigns, one in June 2011 and another, more detailed, in December 2014, the authors attempt to test the hypothesis that "the baseflow in the Chishui forest catchments was not just a mixture of rainwater from different rainfall events, but a mixture of both rainwater and a considerable amount of fog drip water". They conclude that fog in this region contributes between 8% and 31% of baseflow, which at the same time is considered the main component of waterfall discharge. Given the uncertainty in the results, the authors consider this estimate to be the lower bound of fog water contribution to baseflow.

[Figure]

The paper is interesting and the analysis is extensive however, however there are some factors in both format and content that would need to be addressed before the manuscript can be considered for publication in HESS. In terms of format, the structure of the document needs to be improved, for instance, with a clearer and more concise description of the methods both as generic procedures and specific activities for this study, and most importantly a division between results and discussion. In terms of format, although the isotopic analysis is very through, there are several assumptions that add uncertainty to the outcomes, or simply invalidate the results. These are the reasons to recommend major revisions to the document and potential rejection in its current form.

1. Improving the structure and description of the paper: Section 2.2: Consider improving the methods section. This does not mean extending the details and length of this section but being sharper and clearer. L 150-160: The description of the drought is suddenly mixed with the following presentation of methods. L 190-191: For instance here and elsewhere, divide the description of instruments and methods from the description of the monitoring campaigns. Section 3: Divide the section in two: Results, and Discussion separately. This will allow understanding the actual outcomes from the experiments and their implications for understanding the processes and contributions to the system.

2. Improving the content of the paper: L 365-367: Although it is mentioned that isotopes in rainwater changed significantly at the end of the year, without changes in the isotopic composition of stream water, there is a lack of data for the start of the year. To have a clearer picture, it would be necessary to know the stream isotopic composition then as well, to back up some of the assumptions and conclusions of the paper. L 386-388: Although it may seem a bit obvious to agree with the hypothesis that baseflow is a mixture of both rainwater and fog drip, I am afraid there is enough evidence to quantify these contributions. L 398-400: The fact that fog water samples were not collected all year round undermines some of the assumptions and conclusions of the paper. For

instance, what is the seasonality of the fog in the area? What is the isotopic composition of fog depending on such seasonality? Is it correct to assume a constant isotopic composition for the fog? Is it correct to assume a constant isotopic composition in the baseflow and rainwater for the mixing model? Justify these assumptions. L 413-419: The fact that rainfall data comes from monitoring of stations not located specifically in the study area at the local scale necessary to understand the contribution to processes, and especially that rainwater samples were collected at an elevation much lower than the two featured catchments increases substantially the uncertainty in the results. Given the lack of specific data, some of the assumptions, such as considering similar rainfall amounts and isotopic compositions, limit the correct quantification of fog contribution to baseflow. Although the explicit exposition of limitations is very welcome, more specific and detailed data is needed to back up some of the assumptions and conclusions in the study.

3. Minor comments:

L96-97: "It is unclear why frequent fog appears in this region and where the water through the large number of waterfalls in the dry period originates". This is a strong statement. Consider rephrasing and citing relevant sources to support this.

L101: "Using the methods of isotope hydrology" is too of a vague statement. Consider improving the terminology and description of the methods.

L 234: "main stream" should be used instead of "mainstream".

Although the paper is, in general, well written, there are some languages issues that need to be improved to provide a much better presentation of the paper and a clearer exposition of the results.

---

## Author Comment (AC1) · 18 Jun 2018

**Response to anonymous referee #1**

**1 General comments**

The manuscript by Zhan et al. investigates the influence of fog water on the streamflow of streams and water falls in the Chishui River Basin, in particular during the dry season. The investigation relies on the analysis of the isotopic composition of fog, stream, and rain water to calculate the proportional contribution of fog water input to the streamflow.

While the isotope analysis is very thorough and that I think the investigation might be an interesting addition to the literature showing the importance of fog water contribution to the water cycle, I think that a few key aspects of the analysis are presented without any data backing them up, making some of the arguments purely qualitative. As is, I would not recommend this manuscript for publication in HESS.

**Response:** We appreciate the reviewer's constructive comments and suggestions for improvements of the study and manuscript. We share the concern that some arguments are too qualitative. Indeed the statements about the occurrence of fog events were based on the results of previous studies and observations by local residents. Our own observations were made only during the field investigations. Direct data of the fog events were not collected to support these statements. Uncertainties with the quantification of involved hydrological processes and the estimation of fog water contribution to the local water balance also exist.

To further explore these problems (common for hydrological studies ungauged catchments), we managed to acquire additional data of visibility in relation to the fog occurring frequency. These data will be analyzed and added to the revised manuscript. We will also remove "speculative" statements in this regard. Following the comments and suggestions made by both referees, we will restructure the paper to focus on the evidences of fog water's recharge in local water budget, and weaken the discussion about the exact contribution proportion of fog water and underlying mechanism. The detailed responses to the specific comments and corresponding revisions we plan to make are given below.

**2 Specific comments**
**2.1**

Data on the amount and the timing of the fog events is clearly lacking in this study. Currently, fog events are described as happening "often" or "most days". However, in a place with such a strong seasonality in rainfall, one would also expect a seasonality in the fog formation. In particular, data on which days and for how long fog forms can be easily recorded using a simple camera and basic image analysis. I think this information is vital to get a full picture of the hydrology of the site, and this new information could potentially completely change the conclusion.

**Response:** We agree with the suggestions and we are able to fix this problem. Although we did not conduct our own meteorological observation, we have obtained long-term historical visibility data from a local meteorological station. These data will be analyzed and added in the paper to show the frequency and seasonality of fog events.

**Changes to the manuscript:** Analysis of the historical visibility data will be added to the revised manuscript to show the detailed characteristics of fog in the study area.

**2.2**

The geology of the area is invoked throughout the manuscript to explain the type of vegetation present, the occurrence of fog, and the large number of springs and waterfalls. However, no data is provided about the permeability of the different rock formations and details about the underlying soil and rock layers come from general geological information, as opposed to being new information provided by this study. As such, I think the geological argument should be shortened and condensed within the discussion section.

**Response:** The special geology (red sandstone strata) was considered to be an important factor for the formation of the special waterfall landscape in the study area. However, as the reviewer pointed out, some discussions about the relationship between geological condition and local hydrological processes and fog formation lack detailed supporting data. We agree with the reviewer that the geological arguments should be shortened.

**Changes to the manuscript:** Descriptions of the geology in the study site section will be shortened and partly moved to the discussion section. In the discussion part, related geological descriptions will be combined and condensed to make reasonable discussions and assumptions based on the information we already have. Statements with little backup data will be removed.

**2.3**

Organizational issues: I would suggest separating the results and discussion sections. As it is, it is currently very hard to understand how the different arguments work together towards the conclusion. I also think that the subsections in Section 3 could benefit from being further subdivided into subsubsections, making each piece of the argumentation clear. Finally, I am really confused as to why the figures are presented in an order that does not match their order of appearance in the text.

**Response:** Both referees suggested separation of results and discussions. We initially tried to tell the whole story step by step through the analysis of three different sampling campaigns; but it seems hard for readers to follow and understand how the different arguments work together towards the conclusions. We will follow the reviewers' suggestions to restructure the paper to improve its readability.

**Changes to the manuscript:** Results and discussions will be divided into two separate sections. In the results section, general isotopic results from three sampling campaigns will be presented with basic findings drawn. In the discussion part, further analyses, discussions and implications of the results will be presented. The findings from the three sampling campaigns will be combined to reveal the contribution of fog to the hydrological system.

**2.4**

The English language needs to be improved. In the following, I do my best to correct small mistakes and point out sentences that need to be reworked.
Technical issues:
Abstract:
- L19 and L20: "fog", not "fogs"

Introduction:
- L29: dry conditions
- L42: "seasons and thus the..." change to "seasons, resulting in the..."
- L46: may significantly affect
- L48-49: Rephrase the sentence starting with "With more even..."
- L49: has long been assumed to be
- L52 to 55: rephrase the sentence starting with "The contribution of the...."
- L63: measurements, which are
- L64 contributions from fog and rain
- L69: total water input, and was subject
- L89: Maybe replace the beginning of the sentence by: "A question remains as to whether or not fog can..."
- L91: the surrounding area
- L92: also be significantly affected
- L93: the link between fog water and the forest's unique
- L94: the link [...] underlying hydrological processes is not well understood and requires further
- L96-97: rephrase the sentence starting with "It is unclear..."
- L101: remove "Using the methods of isotope hydrology"
Section 2: Materials and Methods
- L140: It is 80m high
- L141: 76.2m wide
- L142: during the rainy season
- L153: remove "affected by this drought event
- L153: the total rainfall from January
- L154-155: which was the lowest record for that period between 1981 and 2015.
- L155: During fieldwork, we observed
- L156: replace "water shortage in stream flow" by "a decrease in stream flow"
- L160: showing little impact of the drought
- L168: A second, more detailed
- L170: streams, waterfalls, and rivers
- L174: during fieldwork
- L176: rephrase or remove "very moist" and "easily seen"
- L179: sandstone cliffs
- L201: in-line
- L202: this technique involves
- L203: the product gases are separated
- L213-216: Please rephrase the sentence starting with: "Data for daily rainfall..."
- L228: isotopic composition
- L254: isotopic composition
- L323: average rate?
- L352: during the dry season
- L353-354: the isotopic composition of the streams in December was similar to that of ... the isotopic composition of the baseflow.

- L363: decreasing trend
- L370: infiltrates and recovers soil saturation
- L377: easy, not easier
- L377: in montane catchments
- L431: an underestimation of the overall fog
- L443: a subtropical humid climate
- L476-478: remove
- L481: into soil together
- L483: intense rainfall. However, most rainwater infiltrates

Figure S1: sampling campaign was conducted from.... Rainfall data from the China Meteorological...

Figure S3: the digital elevation data for Figures S3 to S7, not S2 to S6.

**Response:** We appreciate greatly the reviewer's help on the language issues.

**Changes to the manuscript:** The suggested corrections will be made throughout the revised manuscript and many similar language issues will be fixed.

**2.5**

- L116-132: Remove. See my comment above: the argument based on the geology of the area should be condensed and moved to the discussion.

**Response and changes to the manuscript:** The suggested change will been made.

**2.6**

- L154: Name and exact geolocation of the met station??
- L156: quantify "obvious"
- L161: add details on the brand and the cap/septa type

**Response:** This station is the Xishui meteorological station, located in southeast of the Chishui forest (we described it in the supplementary information). "obvious" was based on our field observation and there was no hydrological data. More details about the sampling vials can be added.

**Changes to the manuscript:** The information of the met station will be added in the main text of the revised manuscript. The statement containing the word "obvious" will be rephrased. Details about the sampling vials including the brand and the cap type will be added.

**2.7**

- L178: what do you mean by the "scale" of a waterfall? Do you mean its size?
- L190: what's an "open site"? Do you mean with full view of the sky (and not under the canopy?)

**Response:** Yes, the "scale" meant the size of waterfall and will be changed to "size". "open site" means site not under the canopy.

**Changes to the manuscript:** These descriptions will be revised to avoid misunderstandings.

**2.8**

- L198-211: You need to give more details about the actual analysis: number of standard used and in which position, post-processing, etc... Also, please explicitly describe the brand, name, and isotope value of the standards used.
- L210: 2permil for 2H on a mass spec is really high. Can you comment on why this value is so high?

**Response and changes to the manuscript:** We agree with your comments on the description of the isotopic measuring methods and instruments. More details about the standard we used for isotopic analysis will be added in the revised manuscript. The precision of $^2$H should be 1‰ for this method.

**2.9**

- L212: Is the LMWL built from the rainfall data you collected and that is cited later on? Or is it from previous studies? Either way, please explain how your LMWL was built.
- L219: What analysis of rainfall-runoff are you referring to?

**Response:** The LMWL was built from the rainfall samples we collected in 2015. The analysis of rainfall-runoff refers to section 3.3.

**Changes to the manuscript:** Related text will be rewritten for clarity.

**2.10**

Section 3: Results and discussion
Estimates of the error in your measurements is missing throughout this entire section.

**Response:** We agree with your concern and more information about the error is needed when showing the results.

**Changes to the manuscript:** Following the reviewer's suggestion, we will add estimations and/or descriptions of errors involved in the results presented.

**2.11**

- Section 3.1: this entire subsection is based on the qualitative analysis of Figure 3. A lot of the conclusions derived are based on two or three data points. Overall, I think that figure 3 and Section 3.1 end up undermining the analysis. Either quantify the "dramatic changes" (L238, L244) or simply remove the figure and section, since I think the following sections/figures are a lot more convincing.

**Responses and changes to the manuscript:** Following the reviewer's suggestions, we are going to rewrite this subsection. Conclusions related to the dramatic changes along the river will be removed. Evidences for the fog water's recharge will be combined with results from other sampling campaigns. The entire results and discussion section will be shortened and reorganized to make the arguments clearer and the conclusions more convincing.

**2.12**

- L273: Maybe Kaseke et al (2017) could be a good, more recent reference to add to this list Kaseke, K. F., Wang, L., & Seely, M. K. (2017). Nonrainfall water origins and formation mechanisms. Science Advances, 3(3), e1603131–. http://doi.org/10.1126/sciadv.1603131

**Response and changes to the manuscript:** The suggested article is very helpful and instructive for our study. This reference will be discussed and cited in the revised

**2.13**

- L286: How about evaporation effects once the water deposits on the leaf?

**Response and changes to the manuscript:** The evaporation on the leaf should also be considered. In the revision, discussions about the evaporation effect on the leaf will be added **.**

**2.14**

- L312: Here would be a good place to have a subsubsection

**Response and changes to the manuscript:** We agree with the suggestion. Results and discussions will be presented separately, and this suggestion will be absorbed.

**2.15**

- L314: "water sourced from precipitation": what water? Soil water, stream water? Please rephrase.
- L326-329: Please rephrase the sentence starting with "the isotopic values of the river..."
- L330-335: Remove, not useful

**Response:** "water sourced from precipitation" includes rain, snow, firn, surface water, and groundwater. These sentences were not well stated and should be rephrased. L330-335 was a short summary for section 3.1 and 3.2, and also introduced the further analysis on the two waterfall-concentrated catchments (Datong and Fengxi) in the following section.

**Changes to the manuscript:** We will rephrase related sentences or paragraphs as suggested.

**2.16**

Section 3.3: here again, there is a real need to provide a more rigorous quantification of the various arguments presented.

- L353: quantify the variation in stable isotope composition during the dry season
- L357: Here, I am not convinced that what is seen in Figure 6 is really a lag in the isotopic composition of stream compared to rain water. In particular, the lack of stream data until May makes it difficult to properly compared both time series. In addition, the difference in sampling time between the rain and the stream water analysis means that you might be missing details in the variation of rain water composition, especially during the rainy summer months.

**Response:** Quantification for some arguments such as the variation of the isotope composition in the stream should be added. However, some details might be missed because of the difference in sampling time between the rain and stream. Therefore, some assumptions and conclusions need a more solid data set, such as the lag in the isotopic composition of stream compared to rain water as well as the detailed hydrological processes we discussed.

**Changes to the manuscript:** Following the reviewer's suggestion, we will remove "speculative" statements that are not well supported by available data. The paper will be revised to focus on the evidence for fog water's recharge to the baseflow, with the

discussions on the rainfall-runoff processes shortened.

**2.17**

- L373: what do you mean by "as the groundwater level"?

**Response and changes to the manuscript:** This sentence will be rewritten for clarity: "As the groundwater level and the rainfall intensity increased from…"

.

**2.18**

- L376: quantify "high permeability

**Response:** The underlying surface is characterized as red sandy soil, resulting from the weathering of red sandstone and considered to have a high permeability.

**Changes to the manuscript:** We will rewritten the concerned sentence for clarity. Discussions and assumptions related with geology will be condensed to be more reasonable and convincing (see our response in 2.2).

**2.19**

- L385: How about the effect of soil water evaporation, which would have a similar effect of enriching rainwater before it enters streams?

**Response and changes to the manuscript:** Normally, the effect of soil water evaporation should affect the isotope composition in the surface runoff. At the study site, if the evaporation effect controls the enrichment, the d-excess of stream water should be smaller than that of the rainwater (i.e. the stream water should be scattered below the LMWL). However, this is not the case as shown in Figure 4, suggesting negligible effect of evaporation in the study area due to the high vegetation coverage and high relative humidity. These points will be added to the revised manuscript.

**2.20**

- L406-445: Here is where additional information on fog events would be really helpful to solidify the argument.
- L428: vary seasonally: data?
- L476: "most days of the year": please show data on fog formation timing!
- L485: fog water is persistent: is it? Again, having fog formation data is crucial to this argument.

**Response and changes to the manuscript:** On these issues related to the detailed data for fog events, please see our responses in 2.1.

**2.21**

- L417: Potential break for a subsubsection

**Response and changes to the manuscript:** Agree. The results and discussion part will be restructured (see response 2.3).

**2.22**

Section 3.5: here is where you could condense all the geological information + include more specific information regarding the different rock formations, in particular permeability.

**Response and changes to the manuscript:** Please refer to our responses in 2.2.

**2.23**

Figure 3: See my comment above, I do not think that this figure helps the analysis and it should be removed or moved to the supplemental information

Figure 4: this is a great summary figure of the isotope data

**Response and changes to the manuscript:** We agree with the suggestions. Both figures will be modified accordingly.

**2.24**

Figure 6: Describe the figure from top to bottom, starting with the rainfall data. Explicitly mention "top", "center", and "bottom". Why is the stream data before May not available?

**Response and changes to the manuscript:** Following the reviewer's suggestion, the discussion about the rainfall-runoff processes will be shortened with related figures (including this one) modified accordingly.

**2.25**

Table 1: it would probably be enough to simply give the average and the STE, and not the range. Add one extra column for the number of samples.

**Response and changes to the manuscript:** Following the reviewer's suggestion, the estimation of fog water's contribution to local water budget will be removed due to the great uncertainties. This paper will be restructured to concentrate on the evidences of fog water's presence in local hydrological system.

**2.26**

Figure S7: Are the differences in wind direction between the 4 maps actually significant?

**Response:** No. As we stated in the figure description, wind mainly blows from the Sichuan Basin and south-eastern areas to the study area over the whole year. Therefore, the Sichuan Basin is considered to be the main water vapour source for the fog formation in the study area.

**Changes to the manuscript:** The description of this figure will be rewritten for clarity.

---

## Author Comment (AC2) · 18 Jun 2018

**Response to anonymous referee #2**

**1 General comments**

This paper analyses isotopic data collected in the Chishui forest region in the southwest of China to characterise the underlying hydrological processes and quantify the contribution of fog water to a waterfall system. By using data collected in two campaigns, one in June 2011 and another, more detailed, in December 2014, the authors attempt to test the hypothesis that "the baseflow in the Chishui forest catchments was not just a mixture of rainwater from different rainfall events, but a mixture of both rainwater and a considerable amount of fog drip water". They conclude that fog in this region contributes between 8% and 31% of baseflow, which at the same time is considered the main component of waterfall discharge. Given the uncertainty in the results, the authors consider this estimate to be the lower bound of fog water contribution to baseflow.

The paper is interesting and the analysis is extensive however, however there are some factors in both format and content that would need to be addressed before the manuscript can be considered for publication in HESS. In terms of format, the structure of the document needs to be improved, for instance, with a clearer and more concise description of the methods both as generic procedures and specific activities for this study, and most importantly a division between results and discussion. In terms of format, although the isotopic analysis is very through, there are several assumptions that add uncertainty to the outcomes, or simply invalidate the results. These are the reasons to recommend major revisions to the document and potential rejection in its current form.

**Response:** We appreciate the reviewer's constructive comments and will revise the manuscript accordingly. We will restructure the paper to concentrate on the evidences of fog water's presence in local hydrological system. Statements that are not supported by available data, especially those on the detailed hydrological processes and the fog water's exact proportion, will be removed.

Detailed responses to the specific comments and possible revisions are listed below.

**2 Specific comments**

**2.1**

Section 2.2: Consider improving the methods section. This does not mean extending the details and length of this section but being sharper and clearer. L 150-160: The description of the drought is suddenly mixed with the following presentation of methods. L 190-191: For instance here and elsewhere, divide the description of instruments and methods from the description of the monitoring campaigns.

**Response:** We agree with your suggestions. The methods section was simply organized in a chronological order of the three sampling campaigns. This part will be restructured following the reviewer's suggestions.

**Changes to the manuscript:** The descriptions of the sampling methods will be rewritten for clarity. A separate paragraph will be added to include our field observing phenomenon and results, as well as the detailed description of the drought in 2011. The description of sampling methods and the instruments we used will also be separated from the description of sampling campaigns.

**2.2**

Section 3: Divide the section in two: Results, and Discussion separately. This will allow understanding the actual outcomes from the experiments and their implications for understanding the processes and contributions to the system.

**Response:** We agree to divide the results and discussion, which is suggested by both two referees.

**Changes to the manuscript:** Results and discussion sections will be separated. In the results section, general isotopic results from three sampling campaigns will be presented with basic findings drawn. In the discussion part, further analyses, discussions and implications of the results will be presented. The findings from the three sampling campaigns will be combined to reveal the contribution of fog to the hydrological system.

**2.3**

L 365-367: Although it is mentioned that isotopes in rainwater changed significantly at the end of the year, without changes in the isotopic composition of stream water, there is a lack of data for the start of the year. To have a clearer picture, it would be necessary to know the stream isotopic composition then as well, to back up some of the assumptions and conclusions of the paper.

**Response and changes to the manuscript:** Similar comments were also made by reviewer 1. We agree with both reviewers – assumptions and conclusions that are not supported by available data should be removed and will do so. The paper will focus on the evidences for fog water's recharge to the baseflow, with the discussions on the rainfall-runoff processes shortened.

**2.4**

L 386-388: Although it may seem a bit obvious to agree with the hypothesis that baseflow is a mixture of both rainwater and fog drip, I am afraid there is enough evidence to quantify these contributions.

L 398-400: The fact that fog water samples were not collected all year round undermines some of the assumptions and conclusions of the paper. For instance, what is the seasonality of the fog in the area? What is the isotopic composition of fog depending on such seasonality? Is it correct to assume a constant isotopic composition for the fog? Is it correct to assume a constant isotopic composition in the baseflow and rainwater for the mixing model? Justify these assumptions.

L 413-419: The fact that rainfall data comes from monitoring of stations not located specifically in the study area at the local scale necessary to understand the contribution to processes, and especially that rainwater samples were collected at an elevation much lower than the two featured catchments increases substantially the uncertainty in the results. Given the lack of specific data, some of the assumptions, such as considering similar rainfall amounts and isotopic compositions, limit the correct quantification of fog contribution to baseflow. Although the explicit exposition of limitations is very welcome, more specific and detailed data is needed to back up some of the assumptions and conclusions in the study.

**Response:** We share the reviewer's concerns with the quantification of fog water's proportion to the baseflow. As we discussed in the manuscript, there were indeed many uncertainties in the estimation. Because fog water was only collected once, the seasonal variations of its isotope composition are unknown, and the exact long-term isotopic input from fog water cannot be determined readily. In addition, the rainwater samples collected at a lower elevation cannot exactly represent the real end member of rainfall input. More long-term measurements are clearly needed to fill these gaps. The quantitative contribution of fog in the stream flow should be determined based on a more solid dataset, which will be established in our future studies.

**Changes to the manuscript:** The estimation of fog water's contribution to local water budget will be removed. The revised paper will concentrate on the evidences of fog water's presence in the local hydrological system.

**2.5**

L96-97: "It is unclear why frequent fog appears in this region and where the water through the large number of waterfalls in the dry period originates". This is a strong statement. Consider rephrasing and citing relevant sources to support this.

**Response:** We are going to add more data (visibility) and references for the occurrence of fogs.

**Changes to the manuscript:** Long-term visibility data in the study area, reflecting the occurrence of fog events, will be added and analyzed. Related statements will be rewritten for clarity.

**2.6**

L101: "Using the methods of isotope hydrology" is too of a vague statement. Consider improving the terminology and description of the methods.

**Response and changes to the manuscript:** Agree. This will be rewritten.

**2.7**

L 234: "main stream" should be used instead of "mainstream".

**Response and changes to the manuscript:** "mainstream" will be replaced by "main stream" throughout the manuscript.

**2.8**

Although the paper is, in general, well written, there are some languages issues that need to be improved to provide a much better presentation of the paper and a clearer exposition of the results.

**Response and changes to the manuscript:** The English writing will be further improved following suggestions of both reviewers'.